# Physical and Thermal Evaluation of Olive Oils from Minor Italian Cultivars

**DOI:** 10.3390/foods10051004

**Published:** 2021-05-04

**Authors:** Maria Paciulli, Graziana Difonzo, Paola Conte, Federica Flamminii, Amalia Piscopo, Emma Chiavaro

**Affiliations:** 1Department of Food and Drug, University of Parma, Parco Area delle Scienze 27/A, 43124 Parma, Italy; emma.chiavaro@unipr.it; 2Department of Soil Plant and Food Sciences, University of Bari “Aldo Moro”, Via Amendola 165/A, 70126 Bari, Italy; graziana.difonzo@uniba.it; 3Department of Agriculture, University of Sassari, Viale Italia 39/A, 07100 Sassari, Italy; pconte@uniss.it; 4Faculty of Bioscience and Technology for Agriculture, Food and Environment, University of Teramo, 64100 Teramo, Italy; fflamminii@unite.it; 5Department of AGRARIA, University Mediterranea of Reggio Calabria, 89124 Reggio Calabria, Italy; amalia.piscopo@unirc.it

**Keywords:** extra virgin olive oil, authenticity, biodiversity, differential scanning calorimetry, color, chlorophyll, harvesting time, geographical origin, botanical origin, principal component analysis

## Abstract

Authentication of extra virgin olive oils is a key strategy for their valorization and a way to preserve olive biodiversity. Physical and thermal analysis have been proposed in this study as fast and green techniques to reach this goal. Thirteen extra virgin olive oils (EVOOs) obtained from minor olive cultivars, harvested at three different ripening stages, in four Italian regions (Abruzzo, Apulia, Sardinia, and Calabria) have been studied. Thermal properties, viscosity and color, as influenced by fatty acid composition and chlorophyll content, have been investigated. The thermal curves of EVOOs, obtained by differential scanning calorimetry, were mostly influenced by the oleic acid content: a direct correlation with the cooling and heating enthalpy and an indirect correlation with the cooling transition range were observed. The minor fatty acids, and particularly arachidic acid, showed an influence, mostly on the heating thermograms. Viscosity and color showed respectively a correlation with fatty acids composition and chlorophyll content, however they didn’t result able to discriminate between the samples. Thanks to the principal component analysis, the most influencing thermal parameters and fatty acids were used to cluster the samples, based on their botanical and geographical origin, resulting instead the harvesting time a less influential variable.

## 1. Introduction

Extra virgin olive oil (EVOO)—which is considered an essential component of the Mediterranean diet, as well as its main source of fats—is appreciated all over the world for its nutritional value and associated health benefits [1,2]. When talking about EVOO, however, it should be considered that there is a wide variety of oils on the market that are often characterized by different quality standards and sensory profiles [3]. In recent years, the increasing demand for olive oil has led to the rapid spread of high-density and super-high-density olive plantations that, although only possible using a limited number of cultivars, maximize productivity and efficiency, providing a more standardized product with an affordable selling price [3,4,5]. At the same time, the main producing countries have vigorously implemented a policy of using as many local cultivars as possible, aiming to preserve olive tree biodiversity and diversify and promote sensory specificity and high-quality local olive oil production [3,5]. This trend has also received the endorsement of the European Union (EU) that, as far back as 1992, introduced the quality trademark Protected Designation of Origin (PDO) to protect and promote typical foods with strong roots in a specific geographic region [6]. Conservation of biological diversity is, in fact, the best tool to ensure species survival, through their adaptability to new environmental conditions and climate change and, in turn, to guarantee long-term sustainability of the entire supply chain [7]. In this scenario, Italy and its very rich olive germplasm—estimated to include about 800 cultivars—play a dominant role, not only in the preservation of olive biodiversity, but also in the production of high-quality olive oils with strong sensory specificity [8,9]. In recent years, several studies have focused on the rediscovery and valorization of minor local Italian cultivars in an attempt to provide valuable genetic resources to be used as strategic elements to increase the sustainability of the future of oil production, pursuing, at the same time, enrichment and diversification of EVOOs to be placed on the market [5,7,9,10,11,12,13].

The authentication of extra virgin olive oils represents a key strategy for their valorization and diversification. Traditionally the traceability of extra virgin olive oils involves their chemical characterization, which is influenced by genotype and different agronomic, environmental, and technological factors [14].

Closely related to chemical composition, but less debated in the literature, is the physical and thermal characterization of olive oils, which could be considered of large interest for consumers and industries.

Differential scanning calorimetry (DSC) has been proposed as an alternative and reproducible method for olive oils identification, through the study of their thermal behavior upon cooling and heating [15]. This technique has been successfully applied in the field of olive oil with the aim to discriminate between commercial categories [16], oxidative status [17], agronomic practices [18], or to detect fraudulent mixtures with other vegetable oils [19]. Some studies have also applied DSC to study the authentication and traceability of extra virgin olive oils by applying chemometric data processing. Chatziantoniou and co-workers [20] successfully determined the botanical origin and geographical origin of six monovarietal extra virgin olive oils originating from four geographical regions of Greece, by applying linear discriminant analysis (LDA) on the data obtained from DSC heating and cooling profiles. DSC in combination with principal component analysis (PCA) was applied to identify EVOO from different Mediterranean countries, revealing how the thermogram obtained upon heating contains important information for sample characterization [21]. An approach based on HPLC-DSC in combination with partial least-square (PLS) regression was used to clarify the influence of triacylglycerol composition on the shape of the cooling curves of EVOOs, to a subsequent authentication of the olive oils [22]. DSC exhibits some advantages compared to the classical analytical methods as it is rapid, does not require sample preparation or solvent utilization, and has a reduced environmental impact.

The measurement of oil viscosity is essential at an industrial level for the selection of proper equipment, such as settling and centrifugation devices, including pumps, pipes, filtration systems, etc. Moreover, from the sensorial point of view, the viscosity can be associated with the term ’fluidity’, where oil with low viscosity means a higher fluidity. Although this subject is not included in the official method [23], the differences perceived between samples can be linked with the oil fatty acid composition [24].

Color is a basic criterion affecting consumer preferences, although the European Union does not require its measurement for the assessment of the virgin olive oil quality. [23] The green shades of olive oil are strictly related to the olive fruit pigments, especially chlorophylls, that are transferred to the oil during the extraction process [25]. Their composition changes during the olive ripening time, influencing both drupe and oils’ color [26]. Olive oil pigments have also been proposed as markers of olive oil’s genetic and environmental make-up [27].

The aim of this work was to analyze thermal profiles and color of thirteen monovarietal extra virgin olive oils, obtained from minor autochthonous Italian olive cultivars, at three different ripening stages, to evaluate the potentiality of these two fast and green methods to differentiate samples based on cultivar-environment-agronomic practice interaction, in relation to the FA and chlorophyll composition.

This approach can be strategic to create a unique and recognizable hallmark for authentication and traceability of extra virgin olive oils from minor Italian cultivars, with the final goal to pursue their valorization and preserve their biodiversity.

## 2. Materials and Methods

### 2.1. Plant Material

Drupes of 13 minor olive Italian cultivars, from 4 Italian regions, were harvested in the 2017 harvest seasons.

The selected cultivars were: Tortiglione (TOR), Dritta (DR) and Gentile dell’Aquila (GEN) from Abruzzo; Sivigliana da olio (SIV), Semidana (SEM) and Corsicana da olio (COR) from Sardinia; Cima di Melfi (CM), Oliva Rossa (OR) and Bambina (BAM) from Apulia; the two clones Ottobratica Cannavà (OTT) and Ottobratica Calipa (OTTC), Tonda di Filogaso (TDF), Ciciarello (CIC) from Calabria.

Temperature data from 2017, in each olive production area, are reported in Table 1.

All the olive trees were located in commercial orchards and grown traditionally. Ten kilograms of drupes were sampled from ten different olive trees, every 15 days (Sampling 1 (t1); Sampling 2 (t2); Sampling 3 (t3)), starting around the middle of October (±one week) at the physiological maturity stage, defined at about 50–70% véraison of the fruits, according to the growers harvesting experience, and as confirmed by the maturity index assessment, as reported by Alamprese et al. [29], on the same olive samples. For each sampling, the drupes were divided into 3 aliquots (around 3 kg each), representing the biological replicates. For each harvesting time, the olive drupes were collected from the same trees and stored at refrigerated temperature overnight, before extraction. VOOs were extracted starting from each cultivar at each harvesting time.

For olive oil extraction, the drupes were milled with a hammer mill. The obtained paste was malaxated at a temperature below 20–25 °C for 30 min and pressed using a hydraulic press (pressure up to 200 bar) in a small olive oil press mill Mini 30 system (Agrimec Valpesana, Firenze, Italy). After centrifugation and filtration through paper, olive oils were then stored in dark glass bottles at room temperature [30].

### 2.2. Fatty Acids Composition

The fatty acid composition was determined after sample transesterification with KOH 2N in methanol [23,31] using a gas-chromatograph system composed of an Agilent Technologies 7890 (Agilent Technologies Inc., Santa Clara, CA, USA), equipped with an FID detector (set at 220 °C) and an SP™ 2340 fused silica capillary column (Supelco, Bellefonte, PA, USA), 60 m length × 0.25 mm i.d. and 0.20 μm film thickness. The temperature of the split injector was 210 °C, with a splitting ratio of 1:100; the detector temperature was 220 °C. The oven temperature was programmed as follows: at the very beginning, the temperature was set at 160 °C then gradually raised to 240 °C. Helium was used as the carrier gas at a flow of 1 mL min^−1^. The identification of each fatty acid was carried out by comparing the retention time with that of the corresponding standard methyl ester (Sigma-Aldrich, St. Louis, MO, USA). The amount of single fatty acids was expressed as area % with respect to the total area [23,31].

### 2.3. Thermal Analysis

EVOO samples (8–10 mg) were weighed in non-hermetic aluminum pans and analyzed by differential scanning calorimetry with a DSC Q100 (TA Instruments, New Castle, DE), following the method of Cerretani et al. [32]. Indium (melting temperature 156.6 °C, ∆Hf = 28.45 J/g) and *n*-dodecane (melting temperature −9.65 °C, ∆Hf = 216.73 J/g) were used to calibrate the instrument and an empty pan was used as reference. Oil samples were equilibrated at 30 °C for 8 min and then cooled at −80 °C at the rate of 2 °C/min, equilibrated at −80 °C for 8 min and then heated from −80 to 30 °C at 2 °C/min. Dry nitrogen was purged in the DSC cell at 50 cm^3^/min. DSC curves were analyzed with Universal Analysis Software (Version 3.9A, TA Instruments) to obtain the enthalpy change for transition (∆H, J/g), onset temperature of transition (Ton,°C), offset temperature of transition (Toff,°C), and peak temperature at the maximum (Tp) for the main events of cooling and heating transitions (p1c and p2c, p1h, p2h, and p3h, °C). The range of transition was calculated as the temperature difference between Ton and Toff for both the cooling and heating transitions.

### 2.4. Viscosity Measurement

Measurements were made using an Advanced Rheometric Expansion System (ARES, Rheometrics (Co)). The viscosity value, in mPas, was calculated on the basis of the speed (100 s^−1^) and the geometry of the probe (Couette cell geometry). Temperature (25 °C) was controlled with a water bath connected to the rheometer. The experiment was carried out using 15 mL of sample. Shear stress was plotted as a function of shear rate using the Orchestrator™^®^ software package and the viscosity (μ) value was obtained from Newton’s law (Equation (1)).
σ = μẏ(1)
where σ is shear stress (mPa), ẏ is the shear rate (1/s) and p is viscosity (mPa s).

### 2.5. Chlorophyll Content

Chlorophylls were determined according to Zago et al. [33]. The chlorophyll content was evaluated by the absorption spectrum according to the American Oil Chemists’ Society [34] and expressed as mg of pheophytin a per kg of oil.

### 2.6. Color

The olive oil color was measured using the software package ImageJ, v.1.38x, fitted with the plugin Color Inspector 3D v. 2.3 [18]. Each time 20 mL of samples were put into a glass Petri dish. The images of each Petri dish were acquired with a scanner (Hewlett Packard, Palo Alto, CA, USA) at 600 dots per inch (dpi). Based on the CIELAB colorimetric system, the measured colorimetric parameters were *L** (lightness); *-a** (green shade); *b** (yellowness).

### 2.7. Statistical Analysis

Means and standard deviations were calculated with the SPSS (version 27.0 SPSS Inc., Chicago, IL, USA) statistical software package. SPSS was used to perform a one-way analysis of variance (ANOVA) and Tukey’s honest significant difference test (HSD) at a 95% confidence level (*p* < 0.05) to identify differences between samples. Pearson correlation coefficients were calculated between the variables at 95% and 99% confidence levels (*p* < 0.05 and *p* < 0.01). Principal component analysis (PCA) was also performed, on normalized data, by means of the Statistica software package (version 8.0, Stat-Soft, Tulsa, OK, USA). PCA was used as a descriptive statistical technique by plotting the normalized independent variables (analytical parameters) versus all cases (samples) with the aim to identify the variables able to discriminate between the cases.

## 3. Results

### 3.1. Fatty Acid Composition

The fatty acid composition is a quality parameter and authenticity indicator of virgin olive oils. The fatty acid composition of the thirteen olive oil samples analyzed in this study is reported in Table 2. Based on these data, all the samples may be classified in the category extra virgin olive oil, according to the European Regulation 2568/91 [23].

The three most abundant fatty acids were oleic (C18:1), palmitic (C16:0), and linoleic (C18:1) acid, as expected. They showed significantly different values between the cultivars. Comparing the samples at t1, C18:1 ranged from 75.5% of TOR and CIC, to 65% of COR; C16:0 ranged from 16.5% to 11% for COR and OR, respectively. C18:2 ranged from 13.5% of SIV and COR to 6.5 of TOR. In general, an opposite trend between C18:1 and C16:0, C18:2 was observed. A clear justification of the observed differences is not that immediate; genetic, environmental, and agronomic factors, alone or in combination, have been reported to influence the composition of olive oils [35]. In particular, the differences in oleic, palmitic, and linoleic acid content seem to be mostly related to the weather: it was reported that lower temperatures could be correlated with a higher content of oleic acid and higher temperatures with a lower content of palmitic and/or linoleic acids [35]. This assumption is partially confirmed by the results of this study (Table 1 and Table 2). The content of some minor fatty acids such as linolenic (C18:3) and arachidic (C20:0) deserves special attention, as their levels are determining factors for the olive oil merceological classification [23]. While C18:3 did not show many differences between the samples, C20:0 showed more variability. In particular, all the cultivars from Abruzzo (TOR, DR, GEN) showed the lowest values (~0.20%), while percentages ranging from 0.45 of OTT to 0.31 of CM were observed for the oils from the other Italian regions.

Considering the differences between the harvesting time, different behaviors were observed between the cultivars. All the Abruzzi cultivars (TOR, DR, GEN), other than CM, OTT, and OTTC, showed an increase in C18:1, passing from t1 to t3. Decreasing values of C18:1 were instead observed for SIV, SEM, TDF, and CIC, from t1 to t3. For the c16:0 content, the opposite was observed in the same cultivars. COR, OR, and BAM did not have differences in oleic acid content, and, moreover, BAM was the most stable cultivar, showing very poor variations of all the fatty acids over time. These trends are also dependent on varietal characters, such as a response to environmental factors, as evidenced by [5,36]. It is reported that higher temperatures during the phases of oil accumulation involve a decrease in oleic acid content [37].

### 3.2. Thermal Analysis

The phase transitions of olive oils measured by DSC are affected by molecular composition changes [38,39]. Figure 1 shows the cooling (A, B, C, D) and heating (E, F, G, H) thermograms of the thirteen studied monovarietal extra virgin olive oils at t1, divided by region of origin. All the curves show common traits: two main transitions upon cooling (p1c, p2c), three main transitions upon heating (p1h, p2h, p3h); analogous thermograms have been already observed for extra virgin olive oils [38,39]. The thermal phenomena observed during cooling are basically influenced by the chemical composition of the samples [38]. In particular, the main exothermic event, peaked at lower temperatures (p1c, Figure 1) has been related to the crystallization of TAG rich in oleic acid. The shape of this transition always appeared as a symmetrical Gaussian curve; suggesting an ordered and cooperative event involving homogenous molecules. The second major exothermic event peak occurred at higher temperatures upon cooling, p2c, and had an asymmetrical shape, indicating the involvement of more heterogeneous molecules, previously identified as saturated triglycerides (TAG) fractions [39].

Intuitively, it might be presumed that the thermograms found upon heating would mirror the ones observed during cooling, in which the formed crystals melt. However, the heating thermograms are more complex. In detail, the first thermal event, p1h, is an exothermic transition, related to a solid-state transformation of the TAG crystals towards more stable forms [40]. p2h and p3h are two endothermic events related to the melting of other TAG polymorphic forms [40]. Bayés-García et al. [40] described the nature of these phenomena very well, by explaining how three main groups of TAGs: triunsaturated OOO and OOL, saturated-unsaturated-unsaturated POO, POL, and SOO, and saturated-saturated-unsaturated PPO, were responsible for the polymorphic behavior observed.

Besides these similarities, each sample showed specific transition temperatures and peak amplitudes and shapes. In some cases, additional thermal events, often visible as shoulders of the main thermal events, both upon cooling and heating, were observed. These minor transitions have not been examined in this study.

Table 3 and Table 4 report the thermal parameters extrapolated from the cooling and heating thermograms, respectively.

Looking at the cooling parameters (Table 3), significant differences between the samples were observed, also showing high correlations with the fatty acid composition. The range of cooling (Range_C), calculated as the difference between Ton_C and Toff_C, at t1 varied from 32.5 °C of TOR, CM, and OR to 42.5 °C of COR. The larger is the cooling transition, the more heterogeneous are the molecules involved in the crystallization [41]. In general, the Apulian and Abruzzi samples had narrower ranges of transition, with lower Ton_C and higher Toff_C, than the Sardinian and Calabrian ones. Narrow cooling transition ranges and low Ton_C have been previously associated with olive oils rich in oleic acid [39], able to perform cooperative crystallization phenomena at lower temperatures. In support of this hypothesis, negative correlations have been found between oleic acid content, Ton_C, and Range_C (*p* < 0.01; R = −0.326; −0.633). The cooling enthalpy (ΔH_C) was also influenced by the oleic acid content. In particular, it was positively correlated with C18:1 (*p* < 0.01; R= 0.397) and negatively with C16:0, C18:2 and C:20 (*p* < 0.01, R= −0.270; −0.392; −0.238). The cooling enthalpy, calculated as the area under the cooling curve, is influenced by the number of molecules involved in the exothermic phenomenon [32]. In this study, at t1, it ranges from 67.5 J/g of TOR and DR, to 62 J/g of SIV. The temperature of the major crystallization peak (Tp1_C) ranged from −42 °C of SIV and COR to −35 °C of BAM. This thermal event, previously associated with the crystallization of oleic rich TAGs [40], in this study, showed positive correlations with C18:1 (*p* < 0.01; R = 0.686) and negative with C16:0, C16:1, C18:2 (*p* < 0.01; R = −0.538; −347; 0.665). The minor exothermal event (p2_c) peaked in a range of temperatures from −9.7 °C of OTT, to −18 °C of OR. Tp2_C correlated positively with C16:0, C16:1, and C20:0 (*p* < 0.01; R = 0.287; 0.401; 0.238) and negatively with C18:1 (*p* < 0.05; R = −0.182) and C18:3 (*p* < 0.01; R = −0.270). These results confirm that p2_c occurs at higher temperatures for oils richer in saturated fatty acids [40].

Fewer differences have been observed comparing the different harvesting times. The parameters that were more affected by olive ripening were Tp1_C and Tp2_C. In detail, Tp1_C increased over time for GEN, CM, OR and decreased for SIV, SEM, and BAM; these trends may be related to the changes of C18:1 observed during ripening (Table 1). Interestingly, for samples not affected by the change in Tp1_C over time, a change in Tp2_C was instead observed. In particular, a decrease in this temperature was observed for TOR, DR, COR, OTT, OTTC, TDF, CIC; it can be related to a decrease in the saturated fatty acids and an increase in the unsaturated ones, with the exception of TDF, CIC for which the opposite trend was observed. Chiavaro et al. [39] observed a significant shift of Ton towards higher temperatures and enlargement of the temperature range as a consequence of ripening on the cooling curves of three monovarietal extra virgin Italian olive oils. These authors suggested an increase in the complexity of oil composition, due to TAG lysis and lipid oxidation. In this study only SIV, OR, and BAM demonstrated a broadening of the crystallization range; however, TDF even showed a narrowing of the transition.

Looking at the heating parameters (Table 4), at t1 significant differences have been observed between the samples. Ton_H ranged from −26.5 °C of CM, OR and CIC, to −37 °C of TOR and DR. Toff_H ranged from 14.5 °C of BAM to 10.5 °C of OR. From these results, it is visible that OR had the narrowest heating transition (Range_H: 37 °C), while TOR, DR, and BAM showed a broader transition (Range_H: 50 °C). Range_H was negatively correlated with C18:0 and C20:0 (*p* < 0.05, R = −465, −0.651); it suggests that the presence of heterogeneous TAGs containing saturated fatty acids formed different polymorphic crystals during the cooling phase, which melt over a wider range of temperatures. The enthalpy of the heating transition ranged from 71.8 J/g of CIC to 64.64 of SIV. This parameter was positively correlated with C18:1 (*p* < 0.01, R = 0.484) and negatively with C16:0 and C18:2 (*p* < 0.01, R = 0.474, 0.431), as already reported in previous studies [18].

Tp1_H ranged from −21.35 °C of SIV to −16 °C of TOR and CIC. Tp2_H ranged from −8 °C of GEN to −4.74 °C of CIC, while Tp3_H ranged from 6 °C of OR to 10.5 °C of BAM. Obtaining a clear correlation of this phenomena with the fatty acid composition is complicated by the kinetic nature of peak p1h and the polymorphisms that characterize p2h and p3h. However, the presence of additional characteristic melting phenomena in the region between peak 1 and peak 2, makes the DSC heating curves of extra virgin olive oil a unique fingerprint for this kind of sample [21].

Comparing the different ripening times, only the olive oil from the cultivar DR did not show any modification. Moreover, Ton_H and ΔH_H were almost stable for all the studied samples. The temperature ranges of the heating transition (Range_H) showed a narrowing tendency for TOR, GEN, SIV, CM, and OTT, as a consequence of the Toff_H shifting towards lower temperatures. Among the three thermal events observed during heating, the first one (p1_h) was exothermic and shifted towards lower temperatures during ripening only for TOR and the three Sardinian cultivars (SIV, SEM, COR). It is not easy to find an explanation of this trend, as it is more related to kinetic phenomena. Tp2_H shifted towards higher temperatures during ripening for TOR, GEN, CM, OR, while it moved to lower temperatures for OTT and OTTC. For these last two samples, it was more clear that this phenomenon could be due to a decrease in the saturated and polyunsaturated fatty acids and an increase in the monounsaturated ones. Tp3_H was the most affected during ripening time; except DR and OR, this thermal event in all other samples shifted towards lower temperatures. These events may be related to the melting of the most saturated TAG polymorphic forms, which tend to decrease over time.

### 3.3. Viscosity

In this study, all the tested olive oils exhibited a linear relationship between shear stress and shear rate, as expected [18,42], allowing olive oil to be classified as a Newtonian fluid. The viscosity of the samples (Figure 2), calculated by Newton’s law (Equation (1)), at t1 ranged from 65.97 mPa*s of SIV to 69.83 mPa*s of TOR, without significant differences between them. These values were in the same order of magnitude as that reported by other authors on virgin olive oils at 25 °C [18,42]. Comparing the viscosity values of the oils obtained from the same cultivar at different ripening times, few differences were measured: COR and OTTC showed, respectively, a decrease and increase in viscosity passing from t1 to t3.

Exploring possible correlations with fatty acids composition, a positive Pearson correlation was found between viscosity and oleic acid (*p* < 0.01; R = 0.276), while an inverse correlation was found with linoleic acid (*p* < 0.01; R = −0.333,). These findings have been already reported by other authors [42,43], as fatty acids with more double bonds, being loosely packed, and exhibiting a more fluid-like behavior.

### 3.4. Chlorophyll Content

Even though the color is not considered a quality attribute in olive oil quality assessment by panel experts [23], consumers use color as a parameter to evaluate olive oil quality and authenticity [44].

The green color of an extra virgin olive oil is due to the presence of chlorophyll; a photosynthetic pigment extracted from olives during milling. During olive ripening, due to catabolic enzymes, chlorophyll undergoes chemical modifications, involving a shift in color from brilliant green to black while going through several shades of purple/pink [25]. This phenomenon, called véraison, literally means a change of color, and is used as an indicator of the ripening stage. Olive farmers start harvesting the olives when they are in the middle of véraison, before full ripeness [13]. The change of color is due to chlorophyll loss and a concomitant increase in anthocyanin pigmentation [45].

It is assumed that the degree of ripening of the olive fruit, and consequently its chlorophyll content, will determine the amount of chlorophyll in the final oil [26].

Figure 3 shows the levels of chlorophyll found in the thirteen olive oils obtained from minor olive Italian cultivars, harvested in three different periods, shifted about two weeks from each other. At time 1 (t1), which represents the optimal olive ripening period, according to the farmers’ experience, the chlorophyll levels ranged from 58.5 mg/kg of BAM, to 5.6 mg/kg of TDF. In general, at t1, the Apulian cultivars (CM, OR, BAM) and SEM, which is a Sardinian cultivar, showed the highest levels of chlorophyll. On the other hand, the Calabrian cultivars (OTT, OTTC, TDF, CIC) had the lowest level of chlorophyll. The amount of chlorophylls in olive oil depends on the olive cultivar, pedoclimatic conditions, and agronomic practices [46].

Comparing the different olives’ harvesting times, in most of the cases the amount of chlorophyll decreased over time. In particular, passing from t1 to t3, the highest chlorophyll loss was registered for CM, which undergoes an 85% loss. Similarly, Criado et al. [47] studied the pigment content in fruit from different olive varieties in six consecutive stages of ripeness. They found that the concentrations of chlorophyll decreased continuously in all the varieties during ripening.

In a few cases, the amount of chlorophyll remained rather constant between t1 and t3 (DR, SEM, OTTC).

### 3.5. Color

As previously reported, the color of olive oil is strictly connected with its pigment content. Confirming this hypothesis, significant correlations between chlorophyll content and colorimetric parameters (Table 5) have been found. In particular, negative Pearson correlations between chlorophylls and the chromatic parameters *L* (*p* < 0.01 R = −0.799) and *a** (*p* < 0.01 R = −0.637) were observed. The higher the chlorophyll content, the darker and greener the olive oil. Surprisingly, the correlation between chlorophylls and the chromatic parameters *b** was positive (*p* < 0.01 R = 0.668). The *b** parameter represents the yellow tones; it was previously related to the carotenoid content in olive oil [47]. We assume that, in this study, the method used for chlorophyll detection measured both chlorophylls *a* and *b*, known to generate intense blue-green and yellow-green shades, respectively [25]. Possibly, the amount of chlorophyll *b* may have influenced this result.

Looking at the differences between the cultivars at t1, the L values ranged from 46–47 of SEM and OR to 56 of DR, resulting in, respectively, the darkest and the lightest samples. Negative *a** values indicate the green color. At t1, OR, and BAM were the greenest samples, with values of −9. On the other hand, DR was the least green sample with values of −6. The color parameter *b** indicates yellow tones; a large variability of this parameter was observed between the samples at t1. The values of *b** ranged from 25 of DR to 56 of COR, resulting in, respectively, olive oils that were less or more yellow.

Focusing on the differences between the harvesting times, for L the general tendency was to increase over time, in relation to the chlorophyll decrease. This phenomenon was especially visible passing from t1 to t2 for most of the cultivars. BAM underwent the highest lightening from t1 to t3 (14%), while DR and GEN did not show any significant change of L. The parameter *a** underwent a general increase from t1 to t3, indicating a progressive loss of greenness. TOR, DR, GEN, and the Apulian cultivar CM did not show significant differences of *a**. On the other hand, the highest loss of greenness was observed for the Calabrian cultivars already at t1. In particular, TDF suffered around a 52% loss of this value prolonging the harvesting time. A large variability between the cultivars was observed in the *b** value trend. A general decrease in the *b** value was observed for the Calabrian cultivars (OTT, OTTC, TDF, CIC), particularly for TDF with a 63.5% loss. On the other hand, the *b** value of the Apulian cultivars (CM, OR, BAM) increased during this time, and DR showed the highest increase in *b** from t1 to t3 (42%). Criado and co-workers [47] observed a decrease in L, an increase in *a**, and a decrease in *b** in two olive oil samples, in relation to the ripening stage of the olive fruit.

### 3.6. PCA

The use of principal component analysis to discriminate between olive oil samples, based on their chemical, physical, and thermal properties, has already been applied successfully [18,41]. In this study, the use of this multivariate statistical technique has been applied to tentatively discriminate between the 13 examined Italian olive oils based on their geographical origin, botanical origin, and olive harvesting time. Starting from twenty-three variables, only nine of them were significant after factor extraction, using an eigenvalue higher than 0.7 as selection criteria. The first two principal components in the PCA accounted for 74.64% of the total variance. Figure 4 shows the projection of the variables on a factor plane.

Most of the selected variables were better described in PC1, which was the most influencing component, able to describe 53.73% of the total variance. Among the fatty acids, C18:1 showed positive factor loadings on PC1, while C16:0 and C18:2 showed negative ones. Considering the thermal properties, heating and cooling enthalpies (ΔH_H, ΔH_C) and the temperature of peak 1 (Tp1_C) showed positive loadings, while the range of cooling (Range_C) showed negative ones. The shift of Tp1_C towards lower temperature and the intensification of ΔH_H and ΔH_C with an increase in C18:1 and an opposite trend with the amount of C16:0 and C18:2 has been already documented [32]. Only two variables were described on PC2, which had a lower influence on the total variance (20.91%). Range_H and C20:0 showed positive and negative values on PC2, respectively. Although Bayés-García and co-workers [40] stated that minor fatty acids do not have an influence on the olive oil thermal transition, in this study, arachidic acid was an influencing parameter for olive oil sample discrimination.

From the factor analysis, the viscosity, chlorophyll content, and all the color parameters were not able to discriminate between the studied olive oils, and, thus, they were excluded from the test.

Figure 5 shows the projection of the cases on a factor plan. It is evident that the main variables influencing clustering were cultivar and geographical of origin, and the harvesting time had less influence. In detail, three main clusters may be distinguished. In the first cluster, the two Sardinian cultivars SIV and COR were well described by the higher concentration of C16:0 and C18:2, with consequent lower enthalpies and values of Tp1_C. The second cluster grouped all the Calabrian cultivars (OTT, OTTC, TDF, CIC), the Apulian cultivars CDM and OLR, together with the Sardinian cultivar SEM; it was characterized by higher levels of C:20, and lower Range H. Within this cluster, the samples CDM, OLR, and OTTC at the latest harvesting stages (t2–t3), shifted towards more positive PC1, indicating the influence of the C18:1 increase during ripening. This phenomenon was even more evident for the three cultivars harvested in Abruzzo (GEN, DR, TOR) for which the t3 largely shifted to the right, indicating an increase in the oleic acid at the last stage of ripening. GEN, DR, and TOR formed together with the Apulian cultivar BAM the third cluster. This cluster, in which a larger distance between the samples was observed, was characterized by high values of both Tp1_C and oleic acid.

Interestingly, also a distribution of the samples according to their region of origin can be observed. In particular, all the Sardinian cultivars showed negative loadings on PC1, while all the Apulian cultivars showed positive loadings on PC1; the Abruzzi and Calabrian cultivars showed, on the other hand, positive and negative loadings on PC2, respectively.

## 4. Conclusions

Thermal analysis is a useful tool to discriminate the EVOOs according to their botanical and geographical origin, which influences the chemical composition of the oil. The Abruzzi cultivars (TOR, DR, GEN), together with BAM, were well differentiated from the others, especially for their higher arachidic acid content, which negatively influenced the range of the heating transition. However, while the Abruzzi cultivars showed an increase in Tp1_C during ripening, BAM showed an opposite trend, in correlation to the oleic acid content. The Sardinian cultivars SIV and COR were mostly characterized by lower values of oleic acid and, consequently, lower transition enthalpies both upon cooling and heating. The Apulian CM, OR, the Calabrian TDF, OTT, OTTC, CIC, and the Sardinian SEM showed similar, intermediate behaviors among others. The olive ripening stage did not particularly influence the olive thermal behavior. The EVOOs’ viscosity and color parameters, despite the correlation with the fatty acid composition and chlorophyll content, respectively, were not selected as a discriminating variables. 

## Figures and Tables

**Figure 1 foods-10-01004-f001:**
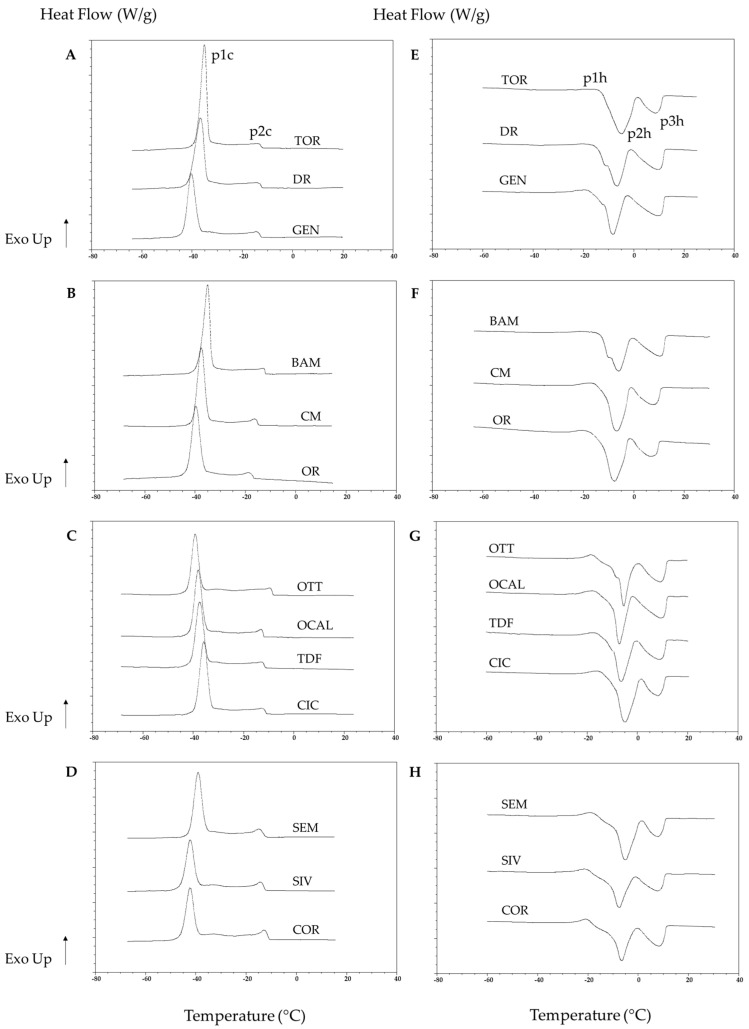
DSC thermograms of the thirteen olive oils (t1) divided by region of origin. **A**–**D**: cooling thermograms; **E**–**H**: heating thermograms. p1c, p2c: main thermal events on cooling; p1h, p2h, p3h: main thermal events on heating.

**Figure 2 foods-10-01004-f002:**
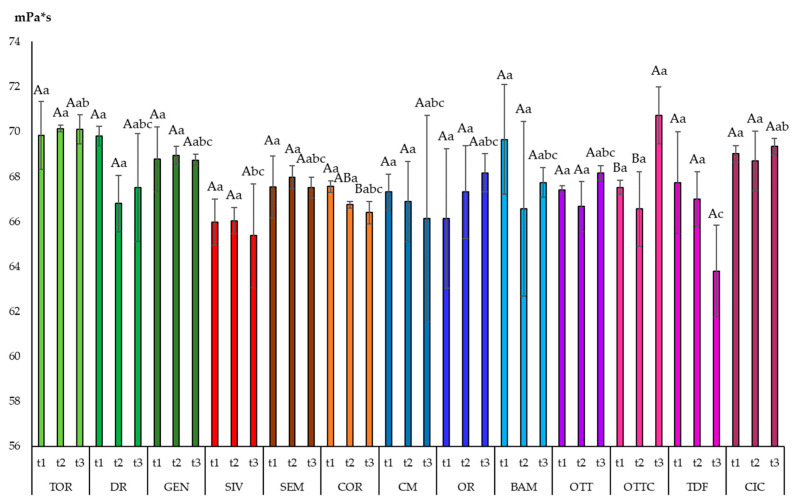
Viscosity of 13 olive oils from minor Italian cultivars harvested at three different maturation stages. Data are expressed as the mean of three replicates ± standard deviations. Different capital letters, between the three harvesting times for the same cultivar, indicate significant differences (*p* < 0.05). Different small letters, at the same harvesting time for the different cultivars, indicate significant differences (*p* < 0.05).

**Figure 3 foods-10-01004-f003:**
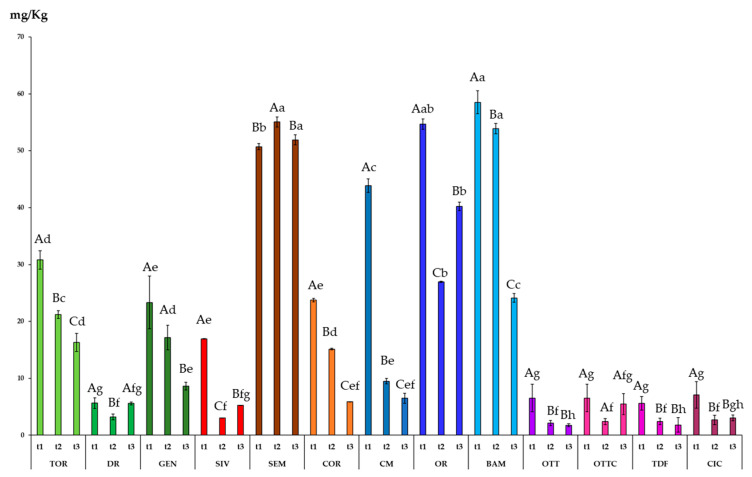
Chlorophyll content of 13 olive oils from minor Italian cultivars harvested at three different maturation stages. Data are expressed as the mean of three replicates ± standard deviations. Different capital letters, between the three harvesting times for the same cultivar, indicate significant differences (*p* < 0.05). Different small letters, at the same harvesting time for the different cultivars, indicate significant differences (*p* < 0.05).

**Figure 4 foods-10-01004-f004:**
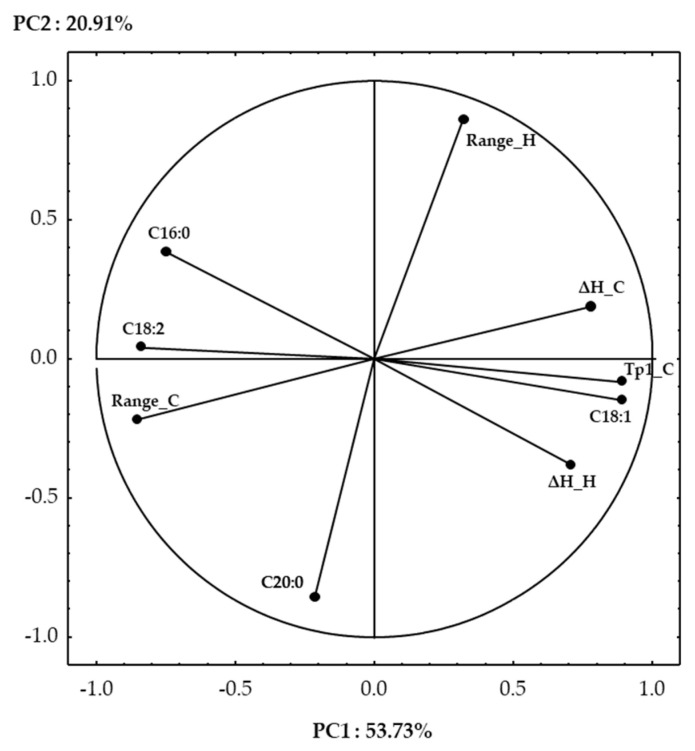
Projections of the variables on a factor plane.

**Figure 5 foods-10-01004-f005:**
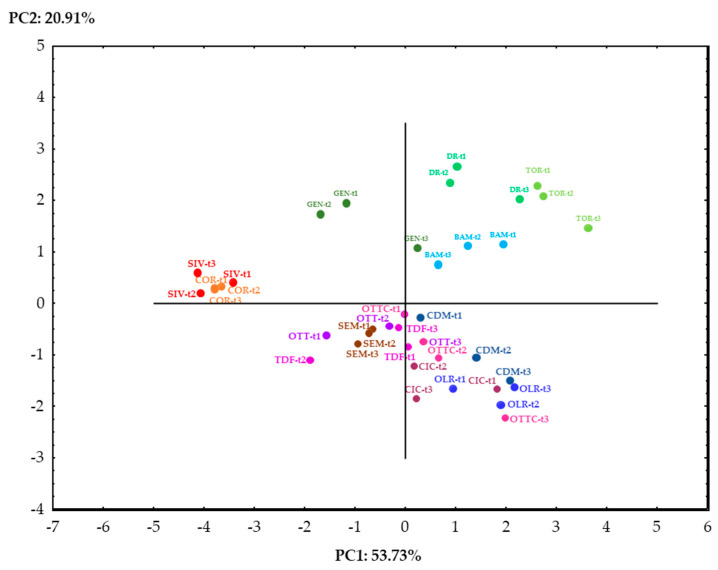
Projections of the cases on a factor plane.

**Table 1 foods-10-01004-t001:** Temperatures (°C) recorded in 2017 in the selected Italian provinces ^1^.

Province	Cultivar	Maximum	Minimum	Average
L’Aquila	TOR, DR	18.0	5.2	11.6
Teramo	GEN	19.9	8.4	14.2
Sassari	SIV, SEM, COR	22.2	11.5	16.9
Bari	CM, OR, BAM	21.4	10.5	16.0
Reggio Calabria	OTT, OTTC, TDF, CIC	23.1	15.9	19.5

^1^ Source: Italian Ministry of Agriculture, Food and Forestry [28]. Tortiglione (TOR), Dritta (DR), Gentile dell’Aquila (GEN), Sivigliana da olio (SIV), Semidana (SEM), Corsicana da olio (COR), Cima di Melfi (CM), Oliva Rossa (OR), Bambina (BAM), Ottobratica Cannavà (OTT), Ottobratica Calipa (OTTC), Tonda di Filogaso (TDF), Ciciarello (CIC).

**Table 2 foods-10-01004-t002:** Fatty acids (%) of 13 olive oils belonging from minor Italian cultivars harvested at three different maturation stages ^1^.

		C_16:0_	C_16:1_	C_18:0_	C_18:1_	C_18:2_	C_18:3_	C_20:0_
	t1	14.05 ± 0.83 ^efA^	0.56 ± 0.03 ^efA^	2.33 ± 0.04 ^bcdeA^	75.75 ± 0.76 ^aB^	6.50 ± 0.08 ^cB^	0.57 ± 0.01 ^abB^	0.24 ± 0.02 ^dB^
TOR	t2	12.41 ± 0.20 ^cB^	0.54 ± 0.00 ^efA^	2.16 ± 0.04 ^deB^	77.13 ± 0.13 ^aA^	6.87 ± 0.04 ^efA^	0.63 ± 0.00 ^bcA^	0.27 ± 0.01 ^eA^
	t3	11.70 ± 0.41 ^efgB^	0.52 ± 0.04 ^fA^	2.23 ± 0.02 ^defB^	77.83 ± 0.43 ^abA^	6.88 ± 0.02 ^fA^	0.57 ± 0.01 ^abB^	0.27 ± 0.00 ^dA^
	t1	15.05 ± 0.43 ^bcdA^	1.59 ± 0.03 ^abA^	1.77 ± 0.05 ^efA^	72.40 ± 0.34 ^bB^	8.25 ± 0.13 ^cB^	0.74 ± 0.08 ^aA^	0.20 ± 0.01 ^dAB^
DR	t2	13.64 ± 0.20 ^abcB^	1.75 ± 0.09 ^bcA^	1.72 ± 0.10 ^fA^	73.08 ± 0.34 ^bcdB^	8.83 ± 0.19 ^cdA^	0.74 ± 0.02 ^aA^	0.23 ± 0.00 ^fA^
	t3	13.19 ± 0.38 ^cdefB^	1.27 ± 0.07 ^cdB^	1.87 ± 0.02 ^fA^	74.84 ± 0.32 ^cdA^	8.07 ± 0.13 ^eB^	0.58 ± 0.22 ^abA^	0.19 ± 0.02 ^eB^
	t1	16.12 ± 0.73 ^abA^	1.96 ± 0.27 ^aA^	1.54 ± 0.02 ^fB^	69.06 ± 0.63 ^cB^	10.46 ± 0.37 ^bAB^	0.63 ± 0.04 ^abA^	0.22 ± 0.01 ^dA^
GEN	t2	16.24 ± 0.30 ^aA^	2.25 ± 0.12 ^aA^	1.55 ± 0.03 ^fB^	67.82 ± 0.94 ^efB^	11.32 ± 0.69 ^bA^	0.61 ± 0.01 ^bcA^	0.20 ± 0.01 ^fB^
	t3	14.66 ± 0.41 ^abcB^	1.99 ± 0.09 ^aA^	2.13 ± 0.10 ^defA^	71.09 ± 0.49 ^fA^	9.34 ± 0.16 ^dB^	0.59 ± 0.02 ^abA^	0.21 ± 0.00 ^eAB^
	t1	15.55 ± 0.07 ^abcB^	1.37 ± 0.01 ^bcdB^	2.49 ± 0.01 ^abcdB^	66.01 ± 0.07 ^deA^	13.74 ± 0.01 ^aC^	0.47 ± 0.01 ^bA^	0.38 ± 0.00 ^abcA^
SIV	t2	15.30 ± 0.01 ^abB^	1.36 ± 0.01 ^cdB^	2.77 ± 0.00 ^bA^	64.50 ± 0.02 ^gB^	15.20 ± 0.03 ^aA^	0.49 ± 0.01 ^efA^	0.38 ± 0.00 ^bcdA^
	t3	16.34 ± 0.28 ^aA^	1.48 ± 0.02 ^bA^	2.40 ± 0.12 ^cdeB^	64.73 ± 0.17 ^hB^	14.24 ± 0.01 ^aB^	0.48 ± 0.02 ^bA^	0.34 ± 0.01 ^cB^
	t1	14.80 ± 0.15 ^bcdA^	0.96 ± 0.01 ^cdeA^	2.57 ± 0.01 ^abcB^	72.12 ± 0.13 ^bA^	8.50 ± 0.02 ^bcC^	0.65 ± 0.01 ^abB^	0.39 ± 0.01 ^abB^
SEM	t2	14.02 ± 0.07 ^abcB^	0.97 ± 0.01 ^deA^	2.61 ± 0.02 ^bcB^	71.98 ± 0.08 ^cdA^	9.31 ± 0.00 ^cB^	0.68 ± 0.00 ^abA^	0.43 ± 0.01 ^aA^
	t3	13.58 ± 0.13 ^cdeC^	0.88 ± 0.01 ^eB^	2.74 ± 0.01 ^abcA^	71.19 ± 0.12 ^fB^	10.54 ± 0.01 ^cA^	0.63 ± 0.01 ^abB^	0.43 ± 0.01 ^bA^
	t1	16.69 ± 0.17 ^aA^	1.41 ± 0.01 ^bcA^	2.42 ± 0.02 ^abcdA^	65.17 ± 0.15 ^eA^	13.35 ± 0.02 ^aC^	0.61 ± 0.01 ^abA^	0.35 ± 0.01 ^bcA^
COR	t2	16.21 ± 0.04 ^aB^	1.30 ± 0.00 ^dB^	2.37 ± 0.00 ^cdeB^	65.14 ± 0.03 ^fgA^	14.03 ± 0.00 ^aB^	0.60 ± 0.01 ^cdA^	0.36 ± 0.00 ^cdA^
	t3	15.92 ± 0.18 ^abB^	1.20 ± 0.02 ^cdC^	2.42 ± 0.02 ^bcdeA^	65.40 ± 0.16 ^hA^	14.16 ± 0.02 ^aA^	0.54 ± 0.00 ^abB^	0.36 ± 0.01 ^cA^
	t1	15.40 ± 0.97 ^abcdA^	0.90 ± 0.40 ^defA^	1.88 ± 0.34 ^defA^	72.44 ± 0.69 ^bB^	8.48 ± 0.51 ^bcA^	0.58 ± 0.02 ^abA^	0.31 ± 0.05 ^cA^
CM	t2	13.72 ± 0.71 ^abcAB^	0.47 ± 0.01 ^fA^	2.37 ± 0.03 ^cdeAB^	74.72 ± 0.55 ^abcA^	7.89 ± 0.14 ^deA^	0.48 ± 0.02 ^efB^	0.35 ± 0.01 ^cdA^
	t3	11.13 ± 1.64 ^gB^	0.48 ± 0.02 ^fA^	2.46 ± 0.02 ^abcdA^	76.24 ± 1.27 ^bcA^	8.73 ± 0.35 ^deA^	0.59 ± 0.02 ^abA^	0.38 ± 0.01 ^cA^
	t1	11.06 ± 0.28 ^gA^	0.49 ± 0.01 ^fB^	2.78 ± 0.02 ^abA^	74.19 ± 0.40 ^abA^	10.46 ± 0.09 ^bA^	0.62 ± 0.26 ^abA^	0.41 ± 0.00 ^abA^
OR	t2	12.10 ± 3.12 ^cA^	0.47 ± 0.03 ^fB^	2.57 ± 0.01 ^bcB^	75.23 ± 2.40 ^abA^	8.50 ± 0.61 ^cdB^	0.75 ± 0.06 ^aA^	0.38 ± 0.03 ^bcA^
	t3	12.13 ± 1.41 ^efgA^	0.56 ± 0.02 ^fA^	2.48 ± 0.03 ^abcdC^	75.32 ± 1.05 ^cdA^	8.44 ± 0.30 ^deB^	0.68 ± 0.05 ^aA^	0.37 ± 0.01 ^cA^
	t1	14.12 ± 0.07 ^defA^	1.46 ± 0.00 ^bA^	1.98 ± 0.01 ^cdefB^	74.01 ± 0.04 ^abA^	7.49 ± 0.03 ^cB^	0.60 ± 0.02 ^abA^	0.35 ± 0.01 ^bcA^
BAM	t2	13.39 ± 0.83 ^bcA^	1.08 ± 0.51 ^dA^	2.32 ± 0.06 ^cdeA^	73.51 ± 1.89 ^bcdA^	8.82 ± 0.58 ^cdA^	0.53 ± 0.04 ^deA^	0.36 ± 0.01 ^cdA^
	t3	13.50 ± 0.11 ^cdeA^	1.33 ± 0.01 ^bcA^	2.27 ± 0.00 ^defA^	72.72 ± 0.13 ^efA^	9.23 ± 0.02 ^dA^	0.59 ± 0.01 ^abA^	0.36 ± 0.01 ^cA^
	t1	16.04 ± 0.22 ^abA^	1.39 ± 0.09 ^bcB^	2.94 ± 0.15 ^aA^	68.22 ± 0.38 ^cdC^	10.44 ± 0.18 ^bA^	0.53 ± 0.01 ^abB^	0.45 ± 0.01 ^aA^
OTT	t2	14.40 ± 0.13 ^abcB^	1.91 ± 0.04 ^abA^	2.07 ± 0.17 ^eB^	71.75 ± 0.71 ^dB^	8.87 ± 0.52 ^cdB^	0.66 ± 0.03 ^bcA^	0.35 ± 0.01 ^dB^
	t3	12.45 ± 0.10 ^defgC^	1.17 ± 0.06 ^dC^	2.79 ± 0.14 ^abcA^	74.31 ± 0.23 ^deA^	8.38 ± 0.18 ^deB^	0.52 ± 0.01 ^abB^	0.36 ± 0.01 ^cB^
	t1	15.04 ± 0.01 ^bcdA^	2.01 ± 0.01 ^aA^	1.97 ± 0.15 ^cdefC^	72.17 ± 0.12 ^bC^	7.77 ± 0.11 ^cA^	0.69 ± 0.01 ^abA^	0.35 ± 0.00 ^bcB^
OTTC	t2	12.69 ± 0.17 ^bcB^	1.12 ± 0.05 ^dB^	2.43 ± 0.11 ^cdB^	76.80 ± 0.39 ^aB^	6.10 ± 0.28 ^fB^	0.45 ± 0.02 ^fB^	0.41 ± 0.01 ^abA^
	t3	11.46 ± 0.13 ^fgC^	0.80 ± 0.11 ^eC^	2.91 ± 0.06 ^aA^	78.46 ± 0.47 ^aA^	5.45 ± 0.30 ^gC^	0.47 ± 0.05 ^bB^	0.46 ± 0.03 ^abA^
	t1	14.44 ± 0.16 ^cdefB^	1.44 ± 0.06 ^bB^	1.62 ± 0.56 ^fB^	73.98 ± 1.04 ^abA^	7.45 ± 0.37 ^cC^	0.68 ± 0.01 ^abA^	0.39 ± 0.01 ^abA^
TDF	t2	15.23 ± 0.15 ^abA^	1.40 ± 0.02 ^cdB^	3.17 ± 0.11 ^aA^	67.69 ± 0.76 ^efC^	11.62 ± 0.51 ^bA^	0.47 ± 0.01 ^efB^	0.41 ± 0.02 ^abA^
	t3	14.35 ± 0.14 ^bcdB^	1.89 ± 0.05 ^aA^	2.00 ± 0.36 ^efB^	71.81 ± 0.69 ^fB^	8.95 ± 0.48 ^deB^	0.65 ± 0.02 ^abA^	0.34 ± 0.00 ^cB^
	t1	13.25 ± 0.26 ^fB^	1.16 ± 0.28 ^bcdA^	2.23 ± 0.27 ^bcdeA^	75.46 ± 2.65 ^aA^	6.96 ± 2.33 ^cB^	0.53 ± 0.01 ^abA^	0.42 ± 0.07 ^abA^
CIC	t2	14.79 ± 0.30 ^abcA^	1.31 ± 0.05 ^cdA^	2.80 ± 0.25 ^bA^	68.47 ± 0.68 ^eB^	11.73 ± 0.59 ^bA^	0.46 ± 0.01 ^efB^	0.44 ± 0.01 ^aA^
	t3	14.83 ± 0.28 ^abcA^	1.28 ± 0.02 ^cdA^	2.86 ± 0.35 ^abA^	68.06 ± 0.45 ^gB^	12.02 ± 0.90 ^bA^	0.46 ± 0.01 ^bB^	0.48 ± 0.03 ^aA^

^1.^ Data are expressed as mean ± standard deviation of three replicates. C18:3 is the sum of alpha and gamma-linolenic acid. ^A, B, C^ in the same column, between the three harvesting times for the same cultivar, indicate significant differences between the means (*p* < 0.05). ^a, b, c, d, e, f, g, h^ in the same column, at the same harvesting time for the different cultivars, indicate significant differences between the means (*p* < 0.05).

**Table 3 foods-10-01004-t003:** DSC cooling parameters of 13 Italian minor olive oils from harvested at three different maturation stages ^1^.

		Ton_C	Toff_C	Range_C	ΔH_C	Tp1_C	Tp2_C
TOR	t1	−11.08 ± 0.51 ^bcdA^	−43.59 ± 0.49 ^aA^	32.51 ± 0.09 ^gA^	67.73 ± 1.64 ^aA^	−35.78 ± 0.62 ^abA^	−13.75 ± 0.17 ^bcA^
	t2	−12.03 ± 0.33 ^eAB^	−43.77 ± 0.67 ^aA^	31.74 ± 1.00 ^fA^	64.45 ± 3.10 ^abA^	−35.09 ± 0.86 ^abA^	−14.99 ± 0.57 ^cdeB^
	t3	−12.82 ± 0.34 ^efB^	−43.47 ± 0.59 ^aA^	30.65 ± 0.82 ^gA^	67.43 ± 2.63 ^abA^	−34.60 ± 0.10 ^abA^	−15.55 ± 0.61 ^efB^
DR	t1	−12.04 ± 0.37 ^deA^	−46.79 ± 0.82 ^bA^	34.75 ± 0.44 ^fgA^	67.12 ± 1.96 ^aAB^	−36.46 ± 0.21 ^bcA^	−14.96 ± 0.67 ^dcA^
	t2	−12.84 ± 0.28 ^fA^	−47.46 ± 0.45 ^bcdeA^	34.62 ± 0.73 ^deA^	65.71 ± 1.02 ^abB^	−36.87 ± 0.37 ^cdeA^	−16.27 ± 0.09 ^efB^
	t3	−12.60 ± 0.41 ^deA^	−46.34 ± 0.99 ^bA^	33.74 ± 1.26 ^fA^	70.45 ± 2.40 ^aA^	−35.56 ± 0.93 ^bA^	−16.90 ± 0.19 ^gB^
GEN	t1	−11.39 ± 0.43 ^bcdA^	−47.48 ± 1.98 ^bcA^	36.09 ± 1.92 ^efA^	64.05 ± 3.20 ^abcA^	−40.19 ± 0.20 ^fB^	−14.00 ± 0.48 ^bcA^
	t2	−12.12 ± 0.33 ^efA^	−49.36 ± 0.34 ^defA^	37.24 ± 0.60 ^cdA^	63.16 ± 1.53 ^abA^	−40.62 ± 0.55 ^iB^	−14.70 ± 0.88 ^cdeA^
	t3	−11.44 ± 0.56 ^cA^	−48.49 ± 0.88 ^cdeA^	37.05 ± 0.56 ^cdeA^	67.04 ± 2.54 ^abcA^	−38.98 ± 0.35 ^cdA^	−13.83 ± 0.37 ^bcA^
SIV	t1	−11.25 ± 0.05 ^bcdAB^	−51.97 ± 0.20 ^efA^	40.72 ± 0.22 ^abC^	61.80 ± 1.85 ^cA^	−42.15 ± 0.13 ^gA^	−14.13 ± 0.44 ^bcA^
	t2	−11.00 ± 0.18 ^bcdA^	−52.70 ± 0.20 ^ghB^	41.70 ± 0.16 ^aB^	61.09 ± 2.91 ^bA^	−43.11 ± 0.24 ^jB^	−14.49 ± 0.53 ^cdeA^
	t3	−11.33 ± 0.12 ^cB^	−53.53 ± 0.03 ^fC^	42.21 ± 0.14 ^aA^	61.49 ± 2.82 ^cA^	−43.27 ± 0.10 ^eB^	−14.93 ± 0.36 ^deA^
SEM	t1	−10.87 ± 0.08 ^bcdA^	−50.29 ± 0.18 ^deA^	39.43 ± 0.26 ^bcdA^	65.02 ± 0.13 ^abcA^	−39.23 ± 0.13 ^efA^	−15.09 ± 0.04 ^cdA^
	t2	−11.42 ± 0.38 ^deA^	−49.82 ± 0.60 ^efA^	38.40 ± 0.49 ^bcA^	63.48 ± 1.38 ^abA^	−39.61 ± 0.16 ^hiB^	−15.18 ± 0.32 ^deA^
	t3	−11.71 ± 0.49 ^cA^	−49.62 ± 0.51 ^eA^	37.91 ± 0.92 ^cdA^	63.73 ± 2.89 ^bcA^	−40.32 ± 0.11 ^dC^	−15.40 ± 0.18 ^eA^
COR	t1	−9.72 ± 0.06 ^abA^	−52.43 ± 0.39 ^fA^	42.71 ± 0.45 ^aA^	65.00 ± 1.47 ^abcA^	−42.44 ± 0.15 ^gA^	−13.25 ± 0.28 ^bA^
	t2	−10.26 ± 0.34 ^bB^	−52.84 ± 0.89 ^ghA^	42.58 ± 1.23 ^aA^	62.96 ± 0.69 ^abB^	−42.59 ± 0.02 ^jAB^	−13.32 ± 0.38 ^bcA^
	t3	−10.41 ± 0.32 ^bB^	−52.58 ± 0.90 ^fA^	42.20 ± 0.99 ^aA^	63.80 ± 0.71 ^bcAB^	−42.70 ± 0.19 ^eB^	−14.25 ± 0.31 ^bcdB^
CM	t1	−13.59 ± 0.11 ^efB^	−47.33 ± 0.21 ^bcA^	33.74 ± 0.31 ^fgA^	64.71 ± 1.02 ^abcA^	−37.58 ± 0.03 ^cdB^	−16.28 ± 0.14 ^deB^
	t2	−12.85 ± 0.04 ^fA^	−47.37 ± 0.59 ^bcdA^	34.53 ± 0.63 ^deA^	65.83 ± 1.20 ^abA^	−35.79 ± 0.18 ^abcA^	−15.12 ± 0.23 ^deA^
	t3	−13.69 ± 0.00 ^fB^	−47.40 ± 0.39 ^bcA^	33.71 ± 0.39 ^fA^	64.51 ± 1.25 ^bcA^	−35.51 ± 0.67 ^bA^	−16.47 ± 0.43 ^fgB^
OR	t1	−14.22 ± 1.90 ^fA^	−47.11 ± 0.10 ^bcA^	32.89 ± 1.80 ^gB^	63.41 ± 0.05 ^abcB^	−38.77 ± 0.95 ^deB^	−17.23 ± 1.61 ^eA^
	t2	−13.70 ± 0.15 ^gA^	−47.20 ± 0.01 ^bcdA^	33.50 ± 0.14 ^efAB^	67.17 ± 0.41 ^aA^	−37.55 ± 0.55 ^defB^	−17.25 ± 0.38 ^fA^
	t3	−11.88 ± 0.22 ^cdA^	−47.63 ± 0.05 ^bcdB^	35.75 ± 0.26 ^defA^	67.31 ± 1.02 ^abA^	−34.21 ± 0.66 ^abA^	−15.04 ± 0.27 ^deA^
BAM	t1	−11.57 ± 0.15 ^cdA^	−44.41 ± 0.53 ^aA^	32.84 ± 0.45 ^gB^	66.79 ± 1.66 ^abA^	−34.83 ± 0.23 ^aA^	−13.30 ± 0.24 ^bA^
	t2	−11.05 ± 0.20 ^cdA^	−45.61 ± 1.26 ^abAB^	34.56 ± 1.27 ^deAB^	65.47 ± 2.25 ^abA^	−36.17 ± 0.66 ^bcdB^	−13.42 ± 0.45 ^bcdA^
	t3	−11.36 ± 0.27 ^cA^	−46.93 ± 0.47 ^bcB^	35.57 ± 0.73 ^efA^	66.02 ± 2.04 ^abcA^	−37.64 ± 0.01 ^cC^	−13.90 ± 0.27 ^bcA^
OTT	t1	−8.31 ± 0.28 ^aA^	−48.94 ± 0.55 ^dA^	40.63 ± 0.83 ^abA^	62.655 ± 0.63 ^bcA^	−38.66 ± 0.61 ^deA^	−9.69 ± 0.05 ^aA^
	t2	−8.81 ± 0.10 ^aAB^	−49.04 ± 0.27 ^cdefA^	40.22 ± 0.33 ^abA^	63.64 ± 1.00 ^abA^	−39.30 ± 0.87 ^ghiA^	−10.33 ± 0.26 ^aAB^
	t3	−9.10 ± 0.34 ^aA^	−49.33 ± 0.37 ^deA^	40.23 ± 0.71 ^abA^	63.72 ± 1.66 ^bcA^	−38.62 ± 0.25 ^cA^	−10.63 ± 0.40 ^aB^
OTTC	t1	−10.51 ± 0.34 ^bcdA^	−48.52 ± 0.43 ^bcdA^	38.01 ± 0.33 ^cdeA^	64.40 ± 1.20 ^abcA^	−37.97 ± 0.10 ^deA^	−12.99 ± 0.13 ^bA^
	t2	−10.57 ± 0.51 ^bcA^	−50.86 ± 1.87 ^fgA^	40.29 ± 2.03 ^abA^	62.44 ± 2.86 ^abA^	−38.47 ± 0.29 ^fghA^	−12.67 ± 0.61 ^bA^
	t3	−11.30 ± 0.09 ^bcA^	−50.20 ± 0.04 ^eA^	38.90 ± 0.05 ^bcA^	65.29 ± 1.68 ^abcA^	−37.62 ± 0.87 ^cA^	−13.65 ± 0.27 ^bB^
TDF	t1	−9.79 ± 0.47 ^abcA^	−50.15 ± 0.49 ^deA^	40.36 ± 0.42 ^abcB^	64.11 ± 1.18 ^abcA^	−37.89 ± 0.58 ^dA^	−12.81 ± 0.30 ^bA^
	t2	−11.09 ± 0.17 ^cdB^	−53.52 ± 1.63 ^hB^	42.43 ± 1.80 ^aA^	63.85 ± 1.23 ^abA^	−37.83 ± 0.23 ^efgA^	−13.28 ± 0.37 ^bcA^
	t3	−11.78 ± 0.17 ^cdB^	−49.81 ± 0.24 ^eA^	38.03 ± 0.07 ^cC^	64.26 ± 0.16 ^bcA^	−37.89 ± 0.25 ^cA^	−14.20 ± 0.18 ^bcdB^
CIC	t1	−10.12 ± 0.26 ^bcA^	−47.34 ± 0.34 ^bcA^	37.23 ± 0.60 ^deA^	66.81 ± 1.33 ^abA^	−35.23 ± 0.86 ^abA^	−13.03 ± 0.21 ^bA^
	t2	−11.13 ± 0.08 ^cdB^	−46.90 ± 0.80 ^bcA^	35.77 ± 0.88 ^cdeA^	67.26 ± 1.73 ^aA^	−34.55 ± 1.34 ^aA^	−13.83 ± 0.12 ^bcdB^
	t3	−11.75 ± 0.33 ^cdC^	−47.12 ± 1.76 ^bcA^	35.37 ± 2.09 ^efA^	67.63 ± 1.57 ^abA^	−33.28 ± 1.38 ^aA^	−14.63 ± 0.04 ^cdeC^

^1.^ Data are expressed as mean ± standard deviation of three replicates. Different capital letters in the same column, between the three harvesting times for the same cultivar, indicate significant differences between the means (*p* < 0.05). Different small letters in the same column, at the same harvesting time for the different cultivars, indicate significant differences between the means (*p* < 0.05).

**Table 4 foods-10-01004-t004:** DSC heating parameters of 13 minor Italian olive oils harvested at three different maturation stages ^1^.

		Ton_H	Toff_H	Range_H	ΔH_H	Tp1_H	Tp2_H	Tp3_H
TOR	t1	−37.29 ± 0.31 ^gA^	13.36 ± 0.05 ^bcdA^	50.65 ± 0.36 ^aA^	69.91 ± 0.53 ^abcdA^	−16.06 ± 0.04 ^aA^	−4.85 ± 0.03 ^abB^	8.59 ± 0.12 ^cdefA^
	t2	−36.96 ± 0.15 ^fA^	12.94 ± 0.34 ^bcdAB^	49.90 ± 0.39 ^aAB^	66.29 ± 2.47 ^deA^	−17.90 ± 1.55 ^bcAB^	−4.71 ± 0.14 ^aB^	8.15 ± 0.05 ^cdB^
	t3	−37.02 ± 0.24 ^fA^	12.47 ± 0.05 ^cdB^	49.49 ± 0.28 ^aB^	70.79 ± 2.58 ^abcA^	−19.27 ± 0.78 ^cdeB^	−4.46 ± 0.10 ^aA^	7.69 ± 0.02 ^deC^
DR	t1	−36.98 ± 0.45 ^gA^	13.59 ± 0.37 ^abcdA^	50.57 ± 0.39 ^aA^	68.83 ± 1.86 ^abcdA^	−20.82 ± 0.32 ^fgA^	−6.80 ± 0.50 ^efgA^	9.29 ± 0.44 ^bcA^
	t2	−36.63 ± 0.34 ^efA^	12.77 ± 0.63 ^cdeA^	49.40 ± 0.97 ^aA^	66.70 ± 0.87 ^deA^	−20.49 ± 1.44 ^fgA^	−7.31 ± 0.64 ^deA^	8.21 ± 0.92 ^cdA^
	t3	−36.30 ± 1.09 ^fA^	13.00 ± 0.25 ^abcA^	49.30 ± 1.22 ^aA^	71.47 ± 3.26 ^abA^	−20.45 ± 0.16 ^defA^	−5.75 ± 0.87 ^bcdA^	8.40 ± 0.28 ^bcA^
GEN	t1	−31.47 ± 0.32 ^eB^	14.10 ± 0.26 ^abA^	45.56 ± 0.27 ^bA^	66.95 ± 2.05 ^deB^	−19.19 ± 0.12 ^deA^	−8.07 ± 0.26 ^iB^	9.79 ± 0.22 ^abA^
	t2	−30.58 ± 0.09 ^dA^	13.53 ± 0.28 ^abAB^	44.11 ± 0.36 ^bB^	68.28 ± 0.68 ^bcdeAB^	−19.89 ± 0.35 ^efA^	−8.28 ± 0.60 ^eB^	9.46 ± 0.12 ^abA^
	t3	−31.03 ± 0.28 ^dAB^	13.14 ± 0.26 ^abB^	44.17 ± 0.54 ^bB^	72.03 ± 2.38 ^abA^	−19.28 ± 1.15 ^cdeA^	−7.09 ± 0.10 ^eA^	8.67 ± 0.07 ^bB^
SIV	t1	−29.86 ± 0.11 ^dA^	11.89 ± 0.06 ^fA^	41.75 ± 0.16 ^cA^	64.64 ± 1.96 ^eA^	−21.35 ± 0.05 ^gA^	−7.70 ± 0.33 ^hiA^	7.65 ± 0.03 ^gA^
	t2	−29.27 ± 0.19 ^cA^	11.17 ± 0.27 ^gB^	40.44 ± 0.43 ^cB^	64.76 ± 2.09 ^eA^	−21.77 ± 0.04 ^gB^	−7.57 ± 0.66 ^eA^	7.28 ± 0.14 ^efB^
	t3	−29.57 ± 0.43 ^cdA^	11.67 ± 0.19 ^efA^	41.24 ± 0.53 ^cAB^	65.52 ± 2.53 ^cA^	−21.84 ± 0.06 ^fB^	−7.03 ± 0.21 ^eA^	7.24 ± 0.08 ^eB^
SEM	t1	−28.45 ± 0.57 ^cA^	12.45 ± 0.09 ^efA^	40.90 ± 0.47 ^cdAB^	67.84 ± 0.88 ^bcdeA^	−19.35 ± 0.08 ^deA^	−5.02 ± 0.03 ^abA^	7.83 ± 0.01 ^efgA^
	t2	−29.26 ± 0.33 ^cA^	12.36 ± 0.11 ^deA^	41.62 ± 0.41 ^cA^	66.63 ± 1.50 ^deA^	−19.61 ± 0.02 ^defB^	−5.50 ± 0.34 ^abAB^	7.65 ± 0.09 ^deB^
	t3	−29.00 ± 0.24 ^bcA^	11.54 ± 0.17 ^fB^	40.54 ± 0.08 ^cB^	67.41 ± 3.18 ^bcA^	−19.80 ± 0.04 ^cdefC^	−5.65 ± 0.15 ^bcB^	7.23 ± 0.08 ^eC^
COR	t1	−28.66 ± 0.18 ^cA^	12.34 ± 0.30 ^efA^	41.00 ± 0.48 ^cdA^	66.42 ± 0.38 ^deA^	−20.85 ± 0.08 ^fgA^	−6.42 ± 0.10 ^defA^	8.36 ± 0.17 ^defgA^
	t2	−29.11 ± 0.36 ^cA^	12.02 ± 0.28 ^efA^	41.12 ± 0.64 ^cA^	66.08 ± 0.42 ^deA^	−21.10 ± 0.01 ^fgB^	−7.30 ± 0.08 ^deB^	7.93 ± 0.08 ^deB^
	t3	−28.90 ± 0.42 ^bcA^	12.23 ± 0.34 ^deA^	41.13 ± 0.56 ^cA^	65.10 ± 0.45 ^cB^	−21.13 ± 0.10 ^efB^	−6.61 ± 0.12 ^deA^	7.68 ± 0.15 ^deB^
CM	t1	−26.46 ± 0.59 ^aA^	12.45 ± 0.18 ^efA^	38.91 ± 0.77 ^fA^	70.83 ± 0.60 ^abA^	−17.46 ± 0.54 ^bA^	−6.76 ± 0.14 ^efgB^	7.74 ± 0.04 ^fgA^
	t2	−26.48 ± 0.56 ^aA^	11.47 ± 0.25 ^fgB^	37.95 ± 0.30 ^dB^	71.78 ± 1.67 ^abcA^	−16.18 ± 0.10 ^aB^	−5.55 ± 0.13 ^abA^	6.61 ± 0.24 ^fB^
	t3	−27.08 ± 0.45 ^aA^	10.74 ± 0.04 ^gC^	37.82 ± 0.40 ^eB^	71.47 ± 1.69 ^abA^	−16.38 ± 0.28 ^abA^	−5.43 ± 0.04 ^bA^	5.99 ± 0.01 ^fC^
OR	t1	−26.40 ± 0.06 ^aAB^	10.62 ± 0.94 ^gB^	37.01 ± 1.00 ^gAB^	70.74 ± 0.01 ^abcC^	−19.03 ± 1.32 ^cdeA^	−7.20 ± 0.65 ^ghB^	6.03 ± 1.34 ^hA^
	t2	−26.08 ± 0.16 ^aA^	10.01 ± 0.06 ^hB^	36.09 ± 0.10 ^eB^	75.57 ± 0.62 ^aA^	−17.48 ± 0.38 ^abcA^	−6.57 ± 0.33 ^cdB^	5.23 ± 0.11 ^gA^
	t3	−26.91 ± 0.62 ^aB^	11.18 ± 0.28 ^fgA^	38.09 ± 0.91 ^deA^	73.43 ± 1.85 ^aB^	−18.01 ± 2.68 ^bcdA^	−4.92 ± 0.57 ^abA^	6.34 ± 0.06 ^fA^
BAM	t1	−35.25 ± 0.34 ^fA^	14.25 ± 0.24 ^aA^	49.50 ± 0.57 ^aA^	69.91 ± 0.78 ^abcdA^	−19.57 ± 0.76 ^efA^	−5.93 ± 0.11 ^cdA^	10.61 ± 0.05 ^aA^
	t2	−35.52 ± 0.98 ^eA^	13.74 ± 0.00 ^aAB^	49.26 ± 0.98 ^aA^	67.15 ± 1.85 ^cdeA^	−20.72 ± 0.11 ^fgA^	−6.19 ± 0.14 ^bcA^	9.72 ± 0.11 ^aB^
	t3	−34.26 ± 1.78 ^eA^	13.29 ± 0.61 ^abB^	47.55 ± 2.39 ^aA^	67.87 ± 0.42 ^abcA^	−18.80 ± 1.80 ^bcdeA^	−6.50 ± 0.52 ^cdeA^	8.73 ± 0.66 ^bC^
OTT	t1	−27.82 ± 0.04 ^bcA^	13.51 ± 0.33 ^abcdA^	41.33 ± 0.37 ^cdA^	67.23 ± 0.07 ^cdeA^	−18.23 ± 0.37 ^bcdA^	−5.56 ± 0.33 ^bcA^	8.98 ± 0.01 ^bcdA^
	t2	−27.37 ± 0.22 ^abA^	12.89 ± 0.28 ^bcdB^	40.26 ± 0.23 ^cAB^	69.14 ± 1.76 ^bcdeA^	−18.66 ± 0.38 ^cdeA^	−6.28 ± 0.26 ^bcB^	8.21 ± 0.22 ^cdB^
	t3	−27.09 ± 0.79 ^aA^	12.69 ± 0.29 ^bcdB^	39.78 ± 1.08 ^cdeB^	67.25 ± 1.51 ^bcA^	−18.38 ± 0.01 ^bcdA^	−6.54 ± 0.22 ^cdeB^	7.98 ± 0.25 ^cdB^
OTTC	t1	−27.85 ± 0.10 ^bcAB^	13.67 ± 0.10 ^abcA^	41.52 ± 0.20 ^cdA^	68.39 ± 0.99 ^abcdA^	−17.84 ± 0.24 ^bcA^	−7.03 ± 0.08 ^fghA^	9.48 ± 0.10 ^bcA^
	t2	−28.05 ± 0.09 ^bcB^	13.40 ± 0.34 ^abcA^	41.45 ± 0.43 ^cA^	66.52 ± 3.10 ^deA^	−17.90 ± 0.44 ^bcA^	−7.71 ± 0.37 ^eB^	8.79 ± 0.30 ^bcB^
	t3	−27.71 ± 0.21 ^abA^	13.35 ± 0.28 ^aA^	41.06 ± 0.49 ^cA^	71.42 ± 2.34 ^abA^	−19.56 ± 1.75 ^cdefA^	−8.16 ± 0.35 ^fB^	9.32 ± 0.33 ^aAB^
TDF	t1	−27.32 ± 0.43 ^abA^	12.99 ± 0.26 ^cdeA^	40.30 ± 0.64 ^deA^	68.17 ± 0.62 ^bcdeA^	−18.28 ± 0.61 ^bcdA^	−6.16 ± 0.41 ^cdeA^	8.72 ± 0.10 ^cdeA^
	t2	−28.02 ± 0.64 ^bcA^	12.61 ± 0.29 ^deAB^	40.62 ± 0.35 ^cA^	69.69 ± 1.46 ^bcdA^	−18.17 ± 0.13 ^bcdA^	−7.36 ± 0.01 ^deB^	7.57 ± 0.25 ^deB^
	t3	−27.83 ± 0.74 ^abcA^	12.26 ± 0.05 ^deB^	40.08 ± 0.79 ^cdA^	69.45 ± 0.69 ^abcA^	−17.78 ± 0.15 ^abcA^	−6.60 ± 0.12 ^cdeA^	7.27 ± 0.13 ^eB^
CIC	t1	−26.54 ± 0.67 ^aA^	12.89 ± 0.15 ^deA^	39.42 ± 0.52 ^efA^	71.80 ± 2.06 ^aA^	−15.70 ± 0.74 ^aA^	−4.75 ± 0.08 ^aA^	8.10 ± 0.00 ^defgA^
	t2	−28.35 ± 1.27 ^bcB^	12.52 ± 0.39 ^deAB^	40.87 ± 1.66 ^cA^	72.70 ± 2.15 ^abA^	−16.47 ± 0.78 ^abA^	−5.31 ± 0.04 ^abB^	7.93 ± 0.20 ^deA^
	t3	−27.32 ± 0.83 ^abAB^	12.38 ± 0.19 ^cdB^	39.69 ± 0.64 ^cdeA^	73.35 ± 1.83 ^aA^	−15.54 ± 0.53 ^aA^	−4.38 ± 0.46 ^aA^	7.37 ± 0.01 ^eB^

^1.^ Data are expressed as the mean ± standard deviation of three replicates. Different capital letters in the same column, between the three harvesting times for the same cultivar, indicate significant differences between the means (*p* < 0.05). Different small letters in the same column, at the same harvesting time for the different cultivars, indicate significant differences between the means (*p* < 0.05).

**Table 5 foods-10-01004-t005:** Color parameters of 13 minor Italian olive oils harvested at three maturation stages ^1^.

		*L*	*a**	*b**
	t1	53.67 ± 0.58 ^Bc^	−7.33 ± 0.58 ^Acd^	53.00 ± 3.46 ^Aab^
TOR	t2	55.00 ± 0.00 ^Acd^	−7.00 ± 0.00 ^Ade^	46.00 ± 2.65 ^ABb^
	t3	55.33 ± 0.58 ^Ac^	−7.00 ± 0.00 ^Ad^	42.33 ± 3.51 ^Bb^
	t1	56.33 ± 0.58 ^Aa^	−6.00 ± 0.00 ^Aa^	25.33 ± 3.21 ^Bd^
DR	t2	56.67 ± 0.58 ^Aab^	−6.00 ± 0.00 ^Acd^	25.00 ± 0.00 ^Bc^
	t3	56.33 ± 0.58 ^Abc^	−6.67 ± 0.58 ^Acd^	36.00 ± 5.20 ^Abc^
	t1	55.67 ± 0.58 ^Aab^	−7.00 ± 0.00 ^Abc^	43.33 ± 8.14 ^Aabc^
GEN	t2	55.00 ± 1.00 ^Acd^	−7.67 ± 0.58 ^Ae^	43.67 ± 4.16 ^Ab^
	t3	56.00 ± 0.00 ^Abc^	−7.33 ± 0.58 ^Ad^	44.33 ± 5.51 ^Ab^
	t1	55.33 ± 0.58 ^Bab^	−8.00 ± 0.00 ^Cd^	46.00 ± 0.00 ^Aabc^
SIV	t2	58.00 ± 0.00 ^Aa^	−6.00 ± 0.58 ^Acd^	26.67 ± 0.58 ^Cc^
	t3	57.67 ± 0.58 ^Aa^	−7.00 ± 0.58 ^Bd^	30.33 ± 0.58 ^Bc^
	t1	46.33 ± 0.58 ^Be^	−8.00 ± 0.58 ^Bd^	50.67 ± 0.58 ^Cabc^
SEM	t2	50.67 ± 0.58 ^Ag^	−7.67 ± 0.58 ^ABe^	53.67 ± 0.58 ^Ba^
	t3	51.00 ± 0.00 ^Ae^	−7.00 ± 0.00 ^Ad^	55.00 ± 0.00 ^Aa^
	t1	55.00 ± 0.00 ^Babc^	−8.00 ± 0.00 ^Bd^	56.00 ± 0.00 ^Aa^
COR	t2	56.00 ± 0.00 ^Abc^	−8.00 ± 0.00 ^Be^	54.00 ± 0.00 ^Ba^
	t3	56.50 ± 0.50 ^Aabc^	−7.00 ± 0.00 ^Ad^	44.00 ± 1.00 ^Cb^
	t1	51.67 ± 0.58 ^Bd^	−7.33 ± 0.58 ^Acd^	55.33 ± 1.15 ^Ba^
CM	t2	56.00 ± 0.00 ^Abc^	−7.00 ± 0.00 ^Ade^	58.67 ± 0.58 ^Aa^
	t3	56.00 ± 0.00 ^Abc^	−7.00 ± 0.00 ^Ad^	59.00 ± 0.00 ^Aa^
	t1	47.00 ± 0.00 ^Ce^	−9.00 ± 0.00 ^Be^	51.00 ± 0.00 ^Cabc^
OR	t2	52.67 ± 0.58 ^Af^	−7.67 ± 0.58 ^Ae^	56.00 ± 0.00 ^Aa^
	t3	49.67 ± 0.58 ^Bf^	−7.00 ± 0.00 ^Ad^	53.67 ± 0.58 ^Ba^
	t1	47.33 ± 0.58 ^Ce^	−9.00 ± 0.00 ^Be^	51.33 ± 0.58 ^Cabc^
BAM	t2	56.00 ± 0.00 ^Abc^	−7.00 ± 0.00 ^Ade^	58.67 ± 0.58 ^Aa^
	t3	54.00 ± 0.00 ^Bd^	−7.33 ± 0.58 ^Ad^	56.33 ± 0.58 ^Ba^
	t1	55.00 ± 0.00 ^Babc^	−6.33 ± 0.58 ^Bab^	39.67 ± 2.52 ^Abc^
OTT	t2	54.33 ± 0.58 ^Bde^	−4.33 ± 0.58 ^Aab^	20.67 ± 2.52 ^Bcd^
	t3	56.50 ± 0.50 ^Aabc^	−4.00 ± 0.00 ^Aab^	18.50 ± 2.50 ^Bd^
	t1	55.00 ± 0.00 ^Babc^	−6.33 ± 0.58 ^Bab^	38.67 ± 10.79 ^Acd^
OTTC	t2	55.00 ± 0.00 ^Bcd^	−5.00 ± 0.00 ^Abc^	26.33 ± 3.21 ^ABc^
	t3	56.00 ± 0.00 ^Abc^	−4.50 ± 0.50 ^Ab^	17.50 ± 0.50 ^Bd^
	t1	54.67 ± 0.58 ^ABbc^	−7.00 ± 0.00 ^Bbc^	43.00 ± 3.00 ^Aabc^
TDF	t2	53.33 ± 0.58 ^Bef^	−3.33 ± 0.58 ^Aa^	15.67 ± 1.15 ^Bd^
	t3	55.67 ± 0.58 ^Abc^	−3.33 ± 0.58 ^Aa^	15.67 ± 0.58 ^Bd^
	t1	54.33 ± 1.15 ^Bbc^	−7.00 ± 0.00 ^Bbc^	48.67 ± 7.77 ^Aabc^
CIC	t2	55.33 ± 0.58 ^ABbcd^	−5.67 ± 0.58 ^Ac^	25.33 ± 4.62 ^Bc^
	t3	56.67 ± 0.58 ^Aab^	−5.67 ± 0.58 ^Ac^	30.3 ± 35.13 ^Bc^

^1.^ Data are expressed as the mean ± standard deviation of three replicates. Different capital letters in the same column, between the three harvesting times for the same cultivar, indicate significant differences between the means (*p* < 0.05). Different small letters in the same column, at the same harvesting time for the different cultivars, indicate significant differences between the means (*p* < 0.05).

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
