# Peer review of "Physical and Thermal Evaluation of Olive Oils from Minor Italian Cultivars"

_foods, 2021, doi:10.3390/foods10051004_

Round 1

Reviewer 1 Report

Comments and suggestions to authors for improvement:

I think it’s an interesting study but there are some mistakes and weakness.

  • Material and methods:
    • 1- I think would be important to know the maturity index of olives (since affect the chemical composition).
    • How many olives were sampled for each variety? For the different sampling time did you harvest the same trees? How about the orchards? Commercial orchards or experimental orchards? Traditional or intensive orchards? We don’t have information.

Also is important to know the olive oil extraction conditions, the trade mark of the olive oil mill, etc.

  • Only one harvesting is not enough. I think more crop seasons would be necessary since there is variability.
  • These results are from 2017 season. Why still are not published?
  • Color in olive oil is due to chlorophyll contribution but also carotenoids. The determination of carotenoids is missing.
  • For physical properties I think also viscosity determination would be important.
  • Why did you use a scanner for color determination instead of a colorimeter?
  • Results:
    • Line 188. In particular differences in oleic acid….seen to be mostly related to the weather.. ? We don´t have information about the weather.
    • The authors described the results in terms of different harvesting times. Anyway the maturity index of the olives could be different depending of the geographical region, weather, …so is not possible to compare different cultivars in different conditions. For this reason the different chlorophyll content, fatty acid content, thermal properties etc could be due to the different maturity degree and not to the cultivar.

Author Response

The authors would like to thank the reviewer for the useful comments. 

he detailed responses are reported in the attached files and the corrections are reported in the text in red colour. 

Best regards. 

Reviewer 2 Report

The manuscript “Physical and thermal evaluation of olive oils from minor Italian cultivars” is generally very well written and contains data of some relevance for a general readers as well as of high relevance for specialists in the topic. Although the subject of the paper could be of interest for the readers of the journal, the paper needs some corrections.

  • Page 2, lines: 70, 78, 81, 84, 89, 90, 93 – Dot should be placed after the parenthesis. Please improve throughout the work.
  • Page 3, line: 123 – Please standardize the degrees of Celsius throughout the work.
  • Materials and Methods; 2.2 Fatty acids composition – Can I ask for the temperature program and injector temperature?
  • Table 1 – Is 18:3 acid alpha or gamma linolenic acid, or the sum of these acids? Please clarify.
  • Page 5, line: 192 – Error in the name and abbreviated formula of arachidic acid. It should be: “arachidic (C20:0)” instead of “arachic (C:20)”.
  • Page 5, line: 192 – Error in the abbreviated formula of oleic acid. It should be: “C18:1” instead of “C:18”.
  • Page 5, lines: 201-201 – For SIV, SEM, TDF and CIC a reduction rather than an increase in the C18:1 acid content was observed.
  • Page 5, line: 202 – Error in the abbreviated formula of palmitic acid. Probably it should be: “C16:0” instead of “C:16”.
  • Page 6, line: 227 – It should be: “exothermic” instead of “exhothermic”.
  • Page 7, line: 262 – Error in the abbreviated formula of arachidic acid. Probably it should be: “C20:0” instead of “C:20”.
  • Page 9, lines: 295 – 297 – “Range_H resulted to be negatively correlated with C18:0 and C20:0 (p<0.05, R= - 465, -0,651); in suggest how the presence of heterogeneous TAGs containing saturated fatty acids influenced the heating transition” -  Could I ask for a more detailed explanation of this statement.
  • Page 14, lines: 420 and 425 – Error in the abbreviated formula of palmitic acid. It should be: “C16:0” instead of “C:16”.
  • Page 14, line: 425 – Replace the round bracket with a square bracket.
  • Page 14, lines: 427 and 444 – Error in the abbreviated formula of arachidic acid. It should be: “C20:0” instead of “C:20”.
  • Throughout the text I propose to replace the word “thermograms” with a word “diagrams” or “curves DSC”.

Author Response

The authors would like to thank the reviewer for the useful comments. 

The detailed responses are reported in the attached file and the corrections are reported in the text in red color. 

Best regards. 

Reviewer 3 Report

Dear Editor,

The work entitled Physical and thermal evaluation of olive oils from minor Italian cultivars, is a work of great importance, since more and more minority olive cultivars have been abandoned and replaced by other cultivars that are more adaptable to the present climatic conditions. . This substitution and abandonment is reflected in the loss of olive oil identity and genetic heritage. The work is well structured, however it needs to be improved in some aspects.
-It should be mentioned how the oils from the different cultivars were extracted and the conditions of extraction from the time of harvest until their extraction.
-It should be mentioned what the maturity index that the cultivars were at different harvest dates.
-In my opinion, a way to relate higher thermal resistance was through the relationship with individual phenols and with vitamin E. The author could also relate using a statistical treatment where he related the oleic acid content with the thermal resistance.

At PCA, perhaps a 3D was able to better show the discrimination between samples. In the way it is presented it is difficult to perceive the discrimination of samples mainly in the 3 and 4 quadrant.

Author Response

The authors would like to thank the reviewer for the useful comments. 

The detailed responses are reported in the attached file and the corrections in the text are reported in red color. 

Best regards. 

Round 2

Reviewer 1 Report

Thanks for the corrections

Reviewer 2 Report

Dear Authors,

Thank you for correcting the paper according to my suggestions.

Best regards

Joanna Bryś

Reviewer 3 Report

Dear Editor,

The work entitled Physical and thermal evaluation of olive oils from minor Italian cultivars, after the suggested changes, presents all the conditions to be published.